# Economics in Marine Spatial Planning: A Review of Issues in British Columbia and Similar Jurisdictions

**Ibrahim Issifu** [1,*], **Ilyass Dahmouni** [1], **Iria García-Lorenzo** [1,2] **and U. Rashid Sumaila** [1]

1 Fisheries Economics Research Unit, Institute for the Oceans and Fisheries, University of British Columbia, Vancouver, BC V6T 1Z4, Canada; ilyass.dahmouni@ubc.ca (I.D.); i.garcia@oceans.ubc.ca (I.G.-L.); r.sumaila@oceans.ubc.ca (U.R.S.)
2 Environmental and Natural Resources Economics Group (ERENEA)–Economics and Business for Society (ECOBAS), Department of Applied Economics, Universidade de Vigo, 36310 Vigo, Spain
* Correspondence: i.issifu@oceans.ubc.ca

**Abstract:** Recently, there has been a rapid increase in the use of Marine Spatial Planning (MSP) worldwide, partly due to the continued loss of marine biodiversity and habitat. The sustainability of marine resources is threatened in all regions of the world by major events such as climate change, marine pollution, and overfishing, as well as illegal, unreported and unregulated fishing both on the high seas and in country waters. Here, we present a comprehensive review and analysis of how economic information has been applied and used to inform decisions about MSP in British Columbia (BC), Canada, and other similar jurisdictions around the world. This focus for the paper was selected because important gaps remain in the literature in terms of incorporating economic questions into MSP. We first present different definitions of MSP, and then we extract useful lessons from MSP regimes with well-tested decision support tools (DSTs) and use this to guide MSP implementation in BC. Finally, we present and discuss case studies from Australia, South Africa, and Belgium. Our review suggests that applying economic information to support the design and implementation of MSPs would lead to better decisions. This in turn would foster livelihoods, attract finance, increase buy-in, and advance United Nations Sustainable Development Goal 14: Life Below Water, thereby achieving Infinity Fish, i.e., ensuring that ocean benefits flow to humanity forever.

**Keywords:** marine spatial planning; decision support tools; economic analysis

## 1. Introduction

Amidst growing threats posed by climate change and pollution in our oceans, the expanding utilization of marine zones for diverse activities such as fishing, transportation, and recreation holds the potential to negatively impact the habitats of marine life, biodiversity, and the vital food security of numerous coastal communities worldwide. Consequently, this issue has ignited a heightened consciousness regarding the need for responsible management of marine areas, both within national jurisdictions and on the high seas [1,2]. Over the past two decades, marine spatial planning (MSP) has become one of the most widely adopted management approaches used to address serious damage to marine areas while improving the well-being of people whose livelihoods are directly linked to these areas. When adopting an MSP strategy, policymakers typically identify a variety of reasons, ranging from environmental and biological to political, socio–economic, and national security.

In this paper, we focus on the economic aspects of the decision-making process and make a twofold contribution to the literature. First, we investigate how economic issues are integrated into MSP efforts in British Columbia (BC), Canada, and other similar jurisdictions. Second, we examine how this economic information has been applied and used to inform MSP decisions. Thus far, there has been little discussion about incorporating economic information in MSP; therefore, our goal is to fill this gap by addressing the following key

questions: (1) Is value-added from MSP analyzed quantitatively? (2) How do you assess the contributions (i.e., jobs, income) of the various sectors of MSP? (3) Are the spatial and temporal allocation of marine scarce resources analyzed? One example of a DST used to analyze the spatial and temporal allocation of marine scarce resources under marine spatial planning is the economic tool of cost-benefit analysis (CBA). The MSP process in the Dutch Exclusive Economic Zone (EEZ) in the North Sea is a notable case of balancing offshore wind farm deployment with the need for sustainable fisheries. This tool was used to assess socio-economic and environmental impacts, offering insights into potential trade-offs [3].

The rest of the paper is organized as follows: Section 2 outlines the general context of the study and definition of MSP. Section 3 introduces the methodology of the study while Section 4 presents a literature review on economic and social perspectives. Section 5 explains how economic questions are incorporated into MSP efforts. Section 6 discusses the results of the review, and Section 7 concludes the paper.

## 2. General Context of the Study and MSP Definition

Marine spatial planning (MSP) is an approach used to manage and allocate the various activities and uses of marine and coastal areas in a systematic and sustainable manner. It involves the spatially and temporally explicit allocation of scarce marine spaces, resources, and services to competing uses, and the governance framework that designs, implements, and monitors these allocations. MSP is employed by the public sector to investigate and distribute human activities in marine areas over time and space to achieve specific objectives set out in policy narratives. The Great Barrier Reef and Canada are recognized for their pioneering efforts in MSP [4].

The Australian government introduced the Marine Park Act 1975 to safeguard coral reefs, while Canada's Oceans Act, passed in 1996, governs the country's national waters. Whereas the latter has generated numerous regulations governing marine protected areas, fishing zones such as marine aquaculture, and the coordination of the territorial sea [5], the former has failed to achieve its objectives, leaving a significant proportion of coral reefs dead. This tragedy is mainly attributable to the bleaching effect caused by rising water temperatures due to climate change [6,7]. MSP aims to respond to increasing pressures on ocean space, support blue growth, preserve important marine zones, resolve conflicts in densely occupied ocean areas, and achieve sustainable development [3,8].

This paper highlights the key features of MSPs, including policy mechanisms, mandates, and regulations, in Canada and comparable jurisdictions. It presents both similarities and distinctions in MSP policies and regulations across six developed countries, as outlined in Table A1 in Appendix A.

MSP can also be defined from an economic perspective as the explicit allocation of scarce marine spaces, resources, and services to competing uses, along with the governance framework that implements and monitors this allocation. It is a process employed by the public sector to investigate and distribute human activities in marine areas over time and space to achieve specific policy objectives. MSP is used by countries such as Vietnam and China to achieve economic and environmental objectives [9]. The concept of MSP has evolved over time, with a focus on it being a continuous effort rather than a one-time action [8]. Various countries articulate their official MSP definitions, as consolidated in Table A2 in Appendix A. The definition of MSP has evolved over time and has been influenced by different stages of its development. Initially, MSP was loosely referred to under various terminologies in the history of fisheries science as elucidated in Table A3 of Appendix A. When applied at an ecosystem level, it represents a practical approach that moves toward ecosystem-based management of marine areas. A seminal study emphasized the importance of assessing the current status of marine areas and the tools that can be used to achieve specific goals, rather than focusing solely on the desired status [9]. It is important to note that marine protected areas (MPAs) and MSPs, while closely related, are distinct concepts with separate definitions. MPAs can support the implementation of MSPs, but they can also be implemented independently to achieve various economic,

biological, and environmental objectives. An MSP is a broader framework that includes MPAs, which in turn includes zoning [10]. Economic assessments have played a significant role in evaluating the value of various marine activities, contributing to the design and management of MPAs by considering the economic costs and benefits associated with conservation measures [3].

Figure 1 shows the three key stages in the use of economics in the MSP process. First, economics provides an understanding of the current state of the stock in terms of biomass, catches, landed values, etc. These bio-economic indicators form powerful tools for assessing the state of the marine zone, particularly in terms of overfishing and environmental degradation. This provides initial crucial information necessary for developing a successful MSP. Next, there are a multitude of DSTs, such as CBA and input–output analysis, which are used to support the development and implementation of the fisheries management strategy. Finally, economics can be used to help steer the MSP towards the desired outcomes.

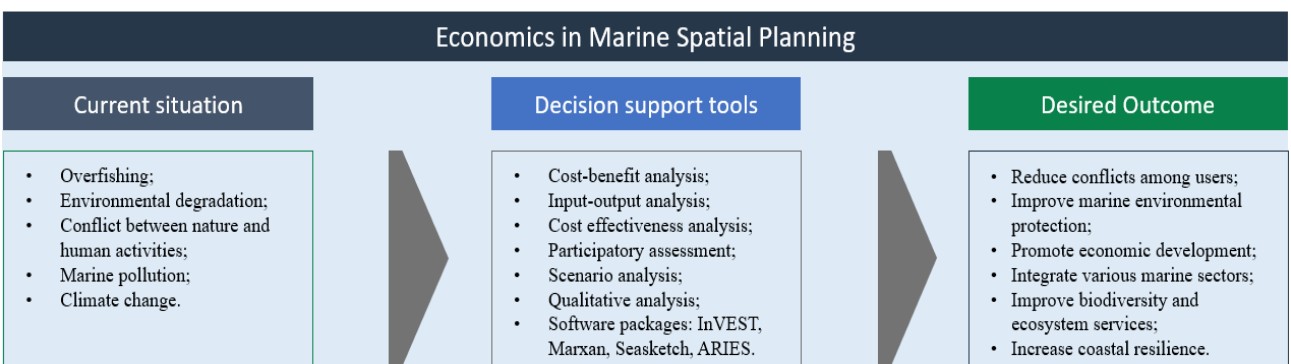

**Figure 1.** Process of using economics in MSP (Source: Authors' creation based on literature reviewed).

### 3. Methodology

Our research aims to describe the questions addressed by economic methods and approaches in the MSP process. This study conducted a comprehensive survey of peer-reviewed and gray literature, focusing on literature directly relevant to the use of economic information to inform MSP decisions. The research focused on jurisdictions similar to British Columbia, Canada, which have initiated or fully implemented MSPs, based on economic development indicators such as gross domestic product (GDP) and the United Nations Human Development Index (HDI). The analysis primarily targeted developed economies and their national jurisdictions. The study did not depict the temporal progression of MSPs but provided a snapshot of available options to date, considering literature published after 2002, taking into account that MSP gained academic recognition in the decades leading up to the year 2000, with the majority of relevant literature emerging post-2000, coinciding with the implementation of the Australian Great Barrier Reef Marine Parks Act 1975.

This review aimed to identify the main socio-economic information relevant to MSP and the range of research questions answered by economic methods and approaches in the selected jurisdictions [3]. The World Bank provides an overview of economic approaches and tools that can strengthen the economic case for MSP. It argues that adding robust economic analysis to the MSP process can increase buy-in, foster livelihoods, attract finance, and advance the long-term blue economy objective of protecting the ocean's resources and ecosystems. This paper emphasizes the spatially and temporally explicit allocation of scarce marine resources and services to competing uses, and the governance framework that designs, implements, and monitors this allocation. It also highlights the importance of understanding the economic trade-offs associated with MSP and the potential of a sustainable ocean economy [3]. The European Maritime Spatial Planning Platform discusses research on the socio-economic aspects of MSP, focusing on methodologies and approaches used in other projects [11].

## 4. Economic and Social Aspects of MSP

### 4.1. Main Socio-Economic Aspects Included in MSP

In a recent study, a strong observed interconnection between ecosystems and marine spatial planning (MSPs) was emphasized, drawing on ecosystem-based management [12–18]. This underscores the pivotal significance of interactions within an ecosystem, not only for MSPs but also for other spatial planning strategies [19].

The integration of human activities into MSPs involves a critical aspect: defining undertakings within the marine environment. Socio-economic factors considered typically include traditional pursuits such as fishing, shipping, transportation, and tourism, alongside emerging sectors such as offshore wind energy, mariculture, dredging, mineral extraction, and biodiversity conservation. MSPs often encompass elements such as land use and zoning, social indicators (e.g., population, age, measures of livelihood sustainability), and economic parameters (employment, monthly income per inhabitant) [20–22].

Over recent decades, MSPs have evolved from simple zoning plans to complex, integrated, adaptive, multiple-use planning [8,23]. Important regulatory changes, such as the European Union (EU) Directive 2014/89/EU on MSP, have significantly shaped these initiatives [24–26].

The European Maritime Spatial Planning Platform highlights the importance of incorporating socio-economic aspects into MSP to reduce or avoid conflicts between economic and non-economic functions and pressures. It also emphasizes the need to consider socio-economic information in MSP data and assessment tools, land–sea interactions, and MSP for blue growth, as well as the challenges and recommendations for integrating socio-economic input into ecosystem-based MSP [22].

The World Bank provides a comprehensive and integrated investment framework for the blue economy through MSP, aiming to reduce investment risk, improve investor certainty, and address environmental and social issues. The World Bank's guidance note Applying Economic Analyses to Marine Spatial Planning offers tools and data for a robust economic analysis of the MSP process, highlighting the potential benefits of such analyses in fostering livelihoods, attracting finance, and drawing financing for marine projects within the blue economy [3].

The literature highlights a challenge in MSP, where socio-economic aspects are not consistently considered, with a predominant focus on environmental dynamics. However, recent developments indicate a growing prominence of social aspects in MSP considerations [27,28]. Concepts such as social sustainability, social equity, social dimensions, and ocean justice are gaining recognition, supported by case studies in various regions [29–38]. Despite the breadth of the subject matter, Table 1 provides examples of the socio-economic aspects addressed in the analyzed case studies.

**Table 1.** Examples of social and economic aspects considered in MSP.

| A. Uses of space, economic activities development, and well-being of coastal communities |
|---|
| Population of the area |
| Activities development in the marine space |
| Gross value added (GVA) by sector of maritime activity |
| Contribution of the sea economy to the GDP |
| GDP/capita of coastal residents |
| Employment rate of coastal population |
| Employment in maritime sectors |
| Poverty, family well-being, gender, health, and education levels |
| Conflicts in the use of maritime space by type and frequency |
| Distribution of the income in the economies onshore |
| Numbers of users of the space (e.g., tourism and recreational activities) |
| Community and citizen participation |

**Table 1.** *Cont.*

| |
|---|
| B. Preservation of social and spiritual values related to ocean |
| Sites and protection of cultural heritage (also underwater heritage) Seascape and landscape Establishment and protection of culturally significant areas for the immaterial cultural values (e.g., definition of cultural values, identification of places of cultural significance, establishment of the relative importance of places of cultural significance) |
| C. Social justice, ocean equity, and social sustainability (recognition, representation, and distribution) |
| Acknowledgment of and respect for pre-existing governance arrangements and history Acknowledgment of and respect for the distinct rights and diversity of needs, worldviews, and lifestyles Access to the resources and benefits distribution (distribution of benefits, risk and harm of decisions, as well as access to resources, with a particular emphasis on vulnerable groups) Equity and fairness of the systems (e.g., equality and inequality changes across communities and groups) Livelihood sufficient, living standards (e.g., education and employment opportunities) Safety and security (e.g., protection against climate change events) Democratic governance and meaningful inclusion of sociocultural values (e.g., participation in decision-making, consideration of individual and group values) |

Source: Summarised by authors based on the case studies of Section 3, especially in [27,28,30,32].

### 4.2. Different Uses of Socio-Economic Aspects in MSP

Social and economic aspects can be included in MSP for different purposes and in different stages, such as planning, development, maturation, and implementation. To include these aspects, multiple strategies can be followed, and DSTs can be used.

The active involvement of all stakeholders in the planning process stands as a central strategy for effectively incorporating socio-economic aspects into MSP, a subject extensively examined [39]. From the engagement of local actors to navigating conflicts among influential entities, including both companies and governments, the convergence of stakeholders presents one of the most intricate challenges for MSP. A Danish case study underscores the importance of understanding power dynamics within MSP, shedding light on how these dynamics shape winners and losers [40]. Through discursive analysis of planning-related documents, regulations, and news reports, the study unveils the mechanisms of power at play and their impact on the MSP process. Notably, the study highlights that those sectors capable of providing more data gain power advantages due to the emphasis on data acquisition and stringent timelines.

Another case study, focusing on the Northeast Ocean Planning process in the US, particularly in Massachusetts Bay, emphasizes the imperative to assess the democratic and inclusive nature of MSP processes to ensure effective participatory engagement [41]. A current study delves into the deeply ingrained traditions and values within planning teams, referred to collectively as planning culture, across three Northern European countries (Denmark, Norway, and Germany), elucidating how planning culture influences MSP [42]. Other case studies further scrutinize stakeholders' social and economic values and perceptions concerning new marine environment situations, such as the introduction of human-made structures [43].

The socio-economic dimensions are not only considered during planning and evaluation but also play a crucial role in monitoring phases. A framework of analysis through the EU-funded MESMA project facilitates monitoring and evaluation in diverse marine areas across Europe [25].

Cross-border resource management poses a unique challenge for MSP, requiring the harmonization of different political, economic, and social perceptions and objectives across multiple jurisdictions. Many studies have explored various aspects of transboundary MSP,

such as defining management authorities, establishing international good practices, and providing evaluation frameworks for institutional integration [9,44–46].

Numerous case studies (including Appendix B) shed light on specific initiatives related to cross-border cooperation, including the SINMORAT project in Spain, Portugal, and France [47], transboundary MSP in the Bay of Biscay [48], and collective action in the Baltic Sea MSP [49]. These initiatives offer insights into challenges and solutions related to social acceptability, ecological and social/management scale mismatches, and joint risk assessments.

A critical aspect of MSP involves securing adequate funding for establishment and management. Economic analysis plays a crucial role in attracting funding, considering long-term policies and financial sustainability throughout the planning, monitoring, evaluation, and revision phases [45]. Funding sources range from government budgets and private sector financing to innovative mechanisms such as blue bonds, foundation funds, and trusts [8]. The importance of aligning funding mechanisms with MSP objectives is emphasized to ensure financial support and sustainable implementation.

Incorporating new activities into marine space, such as marine renewable energies, is a significant challenge addressed by various case studies. These studies analyze factors such as anticipating pressures on ocean energy [50], legal challenges and opportunities [51], institutional barriers, planning priorities in a climate neutrality era [52], and local stakeholder opposition [53].

*4.3. Socio-Economic Aspects in the MSP Decision Support Tools*

In the context of MSP, the effective utilization of DSTs is crucial due to the dynamic nature of maritime space and the diverse interests and conflicts among stakeholders. A comprehensive global review analyzed 34 DSTs employed in 28 MSP initiatives. The findings revealed varying levels of complexity, applicability, and challenges such as limited functionality, tool stability, costs, and the consideration of economic and social decision problems [54]. DSTs should address spatial and temporal dynamics, and be multifunctional, user-friendly, and freely accessible. Geospatial DSTs, utilizing technologies such as GIS/spatial analytics, global navigation satellite systems, earth observation, and others, play a central role in data collection, planning, and environmental monitoring [55].

A study explores the impact of regulatory frameworks, specifically the Directive on MSP, on spatial and non-spatial DST development. The study highlighted the importance of addressing challenges related to uncertainty and incorporating artificial intelligence. [56]. Geospatial technologies, encompassing various tools such as LiDAR, radar, and sonar, are essential for the development of DSTs [55].

DSTs are instrumental in data collection, planning, and monitoring, encompassing socio-economic aspects for detecting areas of specific human uses and pressures [54–56]. They facilitate the development and comparison of alternative scenarios for identifying 'least-cost' solutions and conducting benefit–cost analysis for management measures.

The use of DSTs relies on cumulative impacts assessments (CIA) or cumulative effects assessments (CEA). Challenges include avoiding double counting and addressing complexities and confounding impacts. Innovative models, combining human pressure data and GIS platforms, provide visual representations to support decision-making [57,58].

Various methods, such as the Delphi method, soft systems methodology, and Bayesian modeling, are employed in MSP management [34,42]. Initiatives such as ocean accounting (OA) integrate economic information into MSP, organizing ocean information to support the integrated consideration of social, environmental, and economic values [59].

End-user perspectives on DSTs in MSP emphasize the importance of tool-user interaction and a publicly accepted MSP workflow. Users seek tools with multifunctionality, integrity, and ease of use. Challenges include the underrepresentation of socio-economic information in spatial data, often presented qualitatively and in non-spatial formats [60]. To address these challenges, a participatory mapping approach is proposed [61], collect-

ing stakeholders' knowledge and opinions and translating them into spatial data for a comprehensive representation.

## 5. Application of Economics to Inform MSP Decisions

Examining the potential trade-offs between current and future competing uses of ocean space in monetary terms is integral to effective MSP. Cost and benefit estimation should encompass the full implications of MSP actions, incorporating intrinsic and nonmarket values [62]. Resource allocation in MSP often involves trade-offs, as a portion of marine space allocated to one activity, such as a wind farm, may preclude allocation to other uses, such as oil extraction [63]. Economic desirability, from a social standpoint, is contingent upon overall benefits outweighing costs [63].

Policy-oriented economists utilize models of social conflict to explore how MSP can facilitate win-win situations and resolve conflicts among ocean users. A trade-off analysis focused on alternative ocean uses for the Massachusetts Ocean Management Plan assessed conflicts between offshore wind energy, commercial fishing, and whale-watching sectors, and revealed the potential for preventing losses to the incumbent fishery and whale-watching sectors while generating extra value for the energy sector [16].

Apart from CBA, economic tools such as participatory assessment and spatial software packages such as Integrated Valuation of Environmental Services and Trade-offs (InVEST) 3.9.0 (https://ecosystemsknowledge.net/resources/tool-assessor/invest-integrated-valuation-of-ecosystem-services-and-trade-offs/) are employed to assess MSP's impact on biodiversity and ecosystem services. For instance, in Belize, InVEST was used to analyze trade-offs resulting from alternative Integrated Coastal Zone Management Plan scenarios, aiding decision-making in favor of informed management [64].

Evaluating potential contributions to jobs and incomes generated by various MSP sectors involves using input–output economic tools. A novel study utilized IMPLAN modeling systems to calculate the economic contribution of Washington State's marine sectors, offering insights into the economic impacts of alternative scenarios on both the coastal region and the entire state [65]. Adaptive management, recognized as crucial to MSP evolution, has been effectively employed in countries such as China, Australia, Norway, Germany, Belgium, the Netherlands, and the United States [66,67].

An essential economic question focuses on MSP's role in achieving blue economy goals. An impact pathway assessment helps identify critical policy entry points to reduce agrochemicals or plastic waste in the ocean in Costa Rica [68].

Understanding the true costs and benefits of MSP implementation involves identifying beneficiaries and cost bearers under different scenarios. Considering the differentiated impact on indigenous groups, local communities, and businesses, as shown in Washington's MSP process, helps reduce conflicts and enhances buy-in [65].

Effectively evaluating MSP outcomes is crucial for design and implementation. Causal links between MSP and measurable outcome indicators should be established, considering impacts on poverty dimensions, livelihoods, food security, and wealth generation. Learning from the experience of evaluating MPAs, which has shown economic cases for public investments to improve biodiversity and economic development, can inform MSP evaluation strategies [2,69].

Several economic tools have been identified in the literature as effective in addressing gaps in MSP. These tools aim to integrate economic considerations into the planning process, ensuring the sustainable and inclusive development of marine resources. Some of the key recommendations and case studies from the literature include:

European Maritime Spatial Planning Platform: The platform discusses DSTs in MSP and their present applications, gaps, and future perspectives. It highlights the importance of DSTs in assisting planners with various stages of the MSP process, such as refining goals and objectives, evaluation, and monitoring. The study suggests that future DSTs should consider both spatial and temporal dynamics of the marine environment [48].

The World Bank's MSP Toolkit: The World Bank provides a comprehensive toolkit for MSP, emphasizing the integration of economic considerations to support sustainable and integrated development of economic sectors in healthy oceans. The toolkit includes guidance on applying economic analyses to MSP, which is essential for attracting investment, fostering livelihoods, and improving food security [3].

Several case studies have been conducted to assess the economic impacts linked to MSP using various economic tools, such as input–output techniques. These case studies, such as the German Baltic Sea, Belgium, and the North Sea and the Skagerrak Strait of Norway, provide practical examples of how economic tools have been effectively utilized in the context of MSP [20].

## 6. Discussion

The slow pace of implementing MSP in some coastal jurisdictions can be attributed to various reasons, including financial and economic obstacles, administrative red tape, and the complex nature of MSP as part of a comprehensive jurisdictional strategy [8]. Integrating economic issues into future MSPs presents several challenges, including the need to secure public funds and the consideration of long-term benefits versus short-term costs. Policymakers face the challenge of evaluating MSPs in the context of intergenerational considerations, which raises questions about the weight given to future generations in decision-making processes [62].

Efforts to safeguard the world's oceans are increasingly being emphasized at international meetings, particularly through the United Nations Sustainable Development Goals (SDGs). While SDG 14 (Life Below Water) is directly related to MSP, other goals such as clean water and sanitation, affordable and clean energy, decent work and economic growth, innovation, industry and infrastructure, and climate action are also relevant to MSP. MSP can be driven by SDG 14 and, in turn, can contribute to other goals involving diverse economic components. Additionally, recognizing multi-stakeholder communal use and stewardship of marine space, and the well-being of coastal communities, particularly marginalized groups, presents another challenge [70]. MSP needs to address the viability and sustainability of these communities and their subsistence needs. Economic and social questions have become increasingly relevant in MSP, encompassing not only traditional aspects such as fisheries or transport impacts but also broader issues such as social equity, redistributive capacity, and intangible culture. However, challenges remain, including incorporating long-term change through adaptive approaches and considering transboundary impacts. Technological advances and more participatory and inclusive processes can facilitate addressing these challenges. A better understanding of the costs and benefits, trade-offs, and distributive implications from a socio-economic perspective can guide policymakers, users, and governments in promoting a sustainable ocean economy. The application of economic information and analysis can also ensure that MSP receives sufficient financing for successful implementation [71]. Despite the increasing global adoption of MSP, significant strides are required to bridge existing gaps in incorporating economic dimensions into planning processes.

## 7. Conclusions

This study aimed to evaluate the integration of economic questions into MSP efforts in British Columbia, Canada, and similar jurisdictions. It analyzed the economic tools used by managers to apply economic information and insights to inform MSP decisions. It discussed practical economic questions and tools to incorporate economic information within MSP, highlighting the importance of DSTs in MSP and the attention given to incorporating economic questions. It identified input–output analysis and CBA as the most commonly used methods for estimating the economic impacts of implementing MSP. For example, input–output economic tools are used to estimate the total economic contributions, including indirect and induced impacts of marine sectors. This study demonstrated how each economic tool can be utilized to address potential gaps in MSP while acknowl-

edging the limitations of these tools (such as marine sectoral disaggregation, overestimations, and trade-off estimations) and the need to complement analyses with qualitative approaches. The review emphasized that applying economic information to support the design and implementation of MSPs would lead to better decisions, foster livelihoods, attract finance, and advance the long-term blue economy goals of conserving the oceans' resources and ecosystems.

A notable limitation in the study of the economics of MSP is the challenge of fully integrating and quantifying environmental externalities. While economic analyses often aim to capture the costs and benefits associated with different marine activities, environmental impacts are not always easily translated into monetary terms. This limitation can result in the undervaluation or omission of critical ecological services and the long-term environmental consequences of certain activities. The difficulty in assigning accurate economic values to these externalities may lead to an incomplete understanding of the true economic impact of MSP decisions, potentially undermining the effectiveness of environmental conservation efforts.

Future studies on the economics of MSP could benefit from developing and employing integrated valuation frameworks. These frameworks should aim to comprehensively assess the environmental and economic values of marine resources and activities, including both market and non-market values. By incorporating methods such as contingent valuation, Turnbull estimation, logit/probit estimation, and ecosystem service valuation, researchers can provide a more holistic understanding of the economic and environmental implications of MSP decisions [59]. This approach would allow decision-makers to weigh the trade-offs between different marine uses, considering environmental conservation, social welfare, and economic sustainability in a unified manner.

**Author Contributions:** Conceptualization, I.I., I.D., I.G.-L. and U.R.S.; methodology, I.I., I.D., I.G.-L. and U.R.S.; formal analysis, I.I., I.D., I.G.-L. and U.R.S.; investigation, I.I., I.D., I.G.-L. and U.R.S.; resources, I.I., I.D., I.G.-L. and U.R.S.; writing—original draft preparation, I.I., I.D., I.G.-L. and U.R.S.; writing—review and editing, I.I., I.D., I.G.-L. and U.R.S.; visualization, I.I., I.D., I.G.-L. and U.R.S.; supervision, U.R.S. All authors have read and agreed to the published version of the manuscript.

**Funding:** I.G.-L. was funded by the Xunta de Galicia's Regional, the Spanish Ministry for Science and Innovation and the ERDF (projects ED431C2018/48 and RTI2018-099225-B-100) and also by the Spanish Ministry of Universities under application 33.50.460A.752 and the European Union NextGenerationEU/PRTR through a Margarita Salas contract at the University of Vigo.

**Informed Consent Statement:** Not applicable.

**Data Availability Statement:** Not applicable.

**Acknowledgments:** We thank Policy and Economic Analysis team (DFO Pacific) for their helpful comments.

**Conflicts of Interest:** The authors declare no conflict of interest.

# Appendix A

**Table A1.** Key features of MSPs (e.g., policy mechanism, mandate, regulation, transparency) in Canada and similar jurisdictions (Source: Authors' creation based on literature reviewed).

| | **Canada** | **US** | **UK** | **Belgium** | **Australia** | **Germany** |
|---|---|---|---|---|---|---|
| **National Plan** | **No** | **No** | **No** | **Yes** | **No** | **No** |
| Regional or state plan | Southern BC; Pacific North Coast; Newfoundland & Labrador shelves; Estuary and Gulf of St. Lawrence; and Scotian Shelf and Bay of Fundy. | Northeast Ocean Plan; Massachusetts Ocean Management Plan; Mid–Atlantic Regional Ocean Action Plan; Rhode Island Ocean Special Area Management Plan; Draft Marine Spatial Plan for Washington's Pacific Coast and Oregon Territorial Sea Plan. | East Inshore and East; South Inshore and South Offshore Marine Plans (England); Offshore Marine Plans (England); Scotland's National Marine Plan; Welsh National Marine Plan and Marine Plan for Northern Ireland. | Belgian Part of the North Sea; Flanders. | Great Barrier Reef Marine Park Zoning Plan; Marine Bioregional Plan for the North–west Marine Region; Marine Bioregional Plan for the South–west Marine Region; Marine Bioregional Plan for the North Marine Region; South–east Regional Marine Plan; Marine Bioregional Plan for the Temperate East Marine Region. | Maritime Spatial Plan for the Exclusive Economic Zone (EEZ) of the North Sea; Maritime Spatial Plan for the EEZ of the Baltic Sea; State Development Plan for Schleswig–Holstein; Spatial Planning Program of Lower Saxony; and Spatial Development Program of Mecklenburg–Vorpommer. |
| Responsible ministry at regional plan | Minister of Fisheries and Oceans | Regional Advisory Committees | Marine Management Organization with regional marine management plans | The Belgian Minister for the North Sea | Regional Marine Plans under Commonwealth Environment Department | Ministry of Energy, Infrastructure and Digitalization (Mecklenburg–Vorpommer) |
| Focus area | Environmental protection and economic development | Manage conservation and biodiversity (e.g., Northeast Ocean plan) | Sustainable development | Improve management of the Belgian Part of the North Sea | Support environmental protection and maritime economy | Support maritime economy and environmental protection |
| Legal binding | Ocean Act is non-binding. | Mandated | Not legally binding | Royal Decree | Mandated | General Spatial Planning Act |
| Environment assessment | Cabinet Directive on Strategic Environmental Assessment | | | Strategic Environmental Assessment | | Strategic Environment Assessment |

**Table A2.** Official definition of MSP in different jurisdictions (For each definition, the source is provided in the hyperlink attached to the Authority).

| Country | Authority | Official Terminology | Official Definition |
|---|---|---|---|
| Canada | The Department of Fisheries and Oceans Canada (DFO) | Marine spatial planning | Marine spatial planning is a process for managing ocean spaces to achieve ecological, economic, cultural and social objectives. We advance marine spatial planning in Canada in collaboration with other federal departments, provincial, territorial and Indigenous governments as well as relevant stakeholders. |
| USA | The National Oceanic and Atmospheric Administration (NOAA) | Coastal and marine spatial planning | A compilation of geospatial data to create a national framework for coastal and marine spatial planning, complete with data, tools, and information to bolster transparent, science-based decision-making to enhance regional economic, environmental, social, and cultural well-being. |
| England | Marine Management Organisation | Marine planning | • Marine planning is a new approach to managing the seas around England.<br>• Marine plans guide those who use and regulate the marine area to encourage sustainable development while considering the environment, economy and society. Marine plans apply only in their area, but if a proposed activity may affect the plan area, this should be acknowledged and considered in the application and decision-making. |
| Norway | Government of Norway | Integrated ocean management plans | The purpose of the management plans is to provide a framework for value creation through the sustainable use of marine natural resources and ecosystem services and at the same time maintain the structure, functioning, productivity and diversity of the ecosystems. |
| Belgium, Bulgaria, Croatia, Cyprus, Denmark, Estonia, Finland, France, Germany, Greece, Ireland, Italy, Latvia, Lithuania, Malta, Netherlands, Poland, Portugal, Romania, Slovenia, Spain, Sweden | European Commission (For an update on the adoption of MSP plans in each of the European Commission's member countries, please consult this link) | Marine spatial planning | • MSP is an integrative process to address the increasing demand for maritime space from traditional and emerging sectors while preserving the proper functioning of marine ecosystems. MSP represents a move from traditional single-sector planning to a more integrated approach to the planning of the sea. The key feature of MSP is its functional character i.e., integration of various sectors, societal needs, values, and goals.<br>• MSP can result in plans, permits and other administrative decisions that set the spatial and temporal distribution of relevant existing and future activities and uses in the marine waters, but the outcome of MSP can also take the form of different non-binding visions, strategies, planning concepts, guidelines and governance principles related to the use of sea space. |

**Table A3.** The evolution of MSP.

| Stage | Description | Key Features |
|---|---|---|
| Initial Phase (e.g., Vietnam Danang Master Plan Towards 2030) | – Emerged in the late 20th century as a response to increasing conflicts over marine resource use.<br>– Informal and ad hoc approaches to managing marine space.<br>– Limited integration of environmental considerations. | – Reactive management.<br>– Lack of coordination among stakeholders.<br>– Focus on sector-specific regulations |
| Development Phase (e.g., Ecuador's Galapagos Marine Reserve Management Plan) | – Late 1990s to early 2000s saw the emergence of formal MSP frameworks.<br>– Recognition of the need for holistic and integrated management of marine space.<br>– Growing emphasis on stakeholder engagement and participatory processes. | – Identification of marine planning areas.<br>– Assessment of marine resources and uses.<br>– Integration of ecological, social, and economic considerations.<br>– Stakeholder involvement and public participation. |
| Maturation Phase (e.g., Australia EEZ, including Norfolk Island). | – Mid-2000s onwards witnessed the maturation of MSP as a recognized management approach.<br>– Increasing adoption of MSP at national and regional levels.<br>– Emphasis on ecosystem-based management and sustainability principles. | – Integration of ecosystem-based approach.<br>– Strategic goals and objectives for marine management.<br>– Development of marine spatial plans and zoning systems.<br>– Incorporation of adaptive management and monitoring.<br>– Emphasis on cross-sectoral coordination. |
| Implementation Phase (e.g., E.g., Belize (TS) Integrated Coastal Zone Management Plan, Norway (EEZ and TS) | – Current phase focused on the practical implementation of MSP.<br>– Emphasis on plan implementation, monitoring, and evaluation.<br>– Iterative process to learn and adapt over time.<br>– Continued engagement with stakeholders and adaptive management. | – Implementation of marine spatial plans.<br>– Monitoring of ecological and socio-economic indicators.<br>– Review and revision of plans based on feedback and changing conditions.<br>– Integration with existing governance structures and policies. |

Note: Territorial seas (TS).

**Table A4.** (a) Are trade-offs with monetized values analyzed quantitatively?

| Economic Decision Support Tools (DST) | Scenarios Analysis. |
|---|---|
| Uses of DST | Provides management scenarios and guide management to develop their own solutions. Provides a set of structured information to aid in decision-making and analysis. |
| Country/MSP | USA, European Union, Australia. E.g., scenario planning has been used in Australia for MSP efforts, such as the Great Barrier Reef Marine Park, which considered scenarios of climate change, population growth, and land use changes to guide conservation and management strategies. |
| Limitation | Scenario planning can be resource-intensive, requiring significant time, expertise, and financial investment to develop and analyze. Unreliable data can constrain the effectiveness of the process. |
| Data-intensive level | High: Data on environmental conditions, socio-economic indicators, policy frameworks, and other relevant factors. |
| Output variable | Plausible future scenarios, visualizations of spatial and temporal changes, impact assessments, and identification of key vulnerabilities and opportunities. These outputs inform decision-making by providing a range of potential future outcomes and their associated implications for MSP. |
| References | [72–75] |

Source: Authors' creation based on literature reviewed.

**Table A5.** (b) Does the trade-off analysis consider market and non-market (e.g., ecosystem service value) economic components?

| Economic DSTs | InVEST, Complement with Cost-Effectiveness Analysis |
|---|---|
| Uses of DST | InVEST supports the evaluation of trade-offs between different management options. |
| Country/MSP | China, Costa Rica, Colombia. E.g., InVEST has been employed in China for MSP, including the Beibu Gulf region, to evaluate the impacts of different management scenarios on multiple ecosystem services, such as fisheries production and coastal protection. |
| Limitation | The accuracy of InVEST outputs can be influenced by the scale and resolution of the input data such as ecosystem characteristics, land use, hydrology, and socio-economic factors. |
| Data-intensive level | High: Geospatial data on ecological characteristics, economic indicators, and ecosystem services. |
| Output variable | Spatial maps and models depicting the distribution and value of ecosystem services, trade-off analyses, and valuation of ecosystem services. These outputs contribute to the identification of priority areas for conservation. |
| References | [76–78] |

Source: Authors' creation based on literature reviewed.

**Table A6.** (c) How do you assess the contributions (i.e., jobs, income) of the various sectors of MSP?

| Economic DSTs | Input-Output Analysis |
|---|---|
| Uses of DST | Quantifies the socioeconomic importance of marine sectors in the total economy of a country or region. |
| Country/MSP | German, Norway, Australia, The United States, The United Kingdom. E.g., the German Baltic Sea has utilized input–output analysis in MSP initiatives to assess the economic linkages and impacts of marine sectors such as fisheries, aquaculture, and tourism. |
| Limitation | Input–output analysis assumes a static economic structure, not accounting for innovation, dynamic changes, or future market conditions. Data gaps can affect the accuracy and reliability of the analysis. |
| Data-intensive level | High. Detailed economic data, such as input–output tables, sectoral data on production, employment, and value-added, as well as data on intersectoral linkages and transactions. |
| Output variable | The output variables can include employment impacts, economic multipliers, value-added effects, and the assessment of direct and indirect effects of changes in the marine sector on the overall economy. |
| References | [20,79] |

Source: Authors' creation based on literature reviewed.

**Table A7.** (d) Is value-added from MSP analyzed quantitatively?

| Economic DSTs | The Efficiency Frontier |
|---|---|
| Uses of DST | Allows comparison of very different ecosystem services, for example, MSP increased whale sector values by up to 5% at no cost to the offshore wind energy. |
| Country/MSP | Massachusetts Ocean Management Plan (USA). |
| Limitation | The identification of the efficiency frontier can be influenced by subjective choices, such as the selection of indicators, and weighting factors. |
| Data-intensive level | Medium. Data on resource allocation, management effectiveness, desired outcomes or indicators, and spatial patterns of activities or ecosystem components. |
| Output variable | Outputs may include optimal resource allocation and efficiency scores. The identification of management strategies or spatial plans that achieve desired outcomes most efficiently. |
| References | [16] |

Source: Authors' creation based on literature reviewed.

**Table A8.** (e) How can MSP help to achieve blue economic goals?

| Economic DSTs | Cost-Benefit Analysis (CBA) |
|---|---|
| Uses of DST | CBA provides financial and social rationale for investing in the BE. |
| Country/MSP | CBA has been applied in MSP projects. In Indonesia, NPV of an Indonesian MPA was estimated as USD 3.55.0 million. |
| Limitation | Social and ecological aspects may be challenging to monetize. |
| Data-intensive level | Economic data, market prices, valuation techniques. |
| Output variable | Net present value (NPV), benefit-cost ratio, economic impact assessments. |
| References | [3,80] |

Source: Authors' creation based on literature reviewed.

**Table A9.** (f) What are optimal conservation areas for MPA designs?

| Economic DST | Marxan |
|---|---|
| Uses of DST | Designing MPAs and conservation planning. |
| Country/MSP | Marxan applied in the Great Barrier Reef Marine Park (Australia) to identify MPA designs that maximize conservation outcomes. |
| Limitation | Computationally intensive, sensitive to input parameters, and may not account for all stakeholder preferences. |
| Data-intensive level | High: Biodiversity data, habitat maps, socioeconomic data. |
| Output variable | Optimal MPA configurations, conservation priorities, and cost-effectiveness analysis. |
| References | [81] |

Source: Authors' creation based on literature reviewed.

**Table A10.** (g) How do you integrate economic information or various data layers to inform MSP decision-making?

| Economic DST | Geographic Information Systems (GIS) |
|---|---|
| Uses of DST | Spatial data management, visualization, and analysis. |
| Country/MSP | GIS used in the MSP process of the United Kingdom's Marine Management Organization to integrate various data layers and inform decision-making. |
| Limitation | GIS requires georeferenced data, expertise in GIS software 3.34.2, and data quality control. |
| Data-intensive level | Geospatial data on human activities, marine features, and habitats. |
| Output variable | Maps, spatial analysis results, and visualization. |
| References | [11] |

Source: Authors' creation based on literature reviewed.

**Table A11.** (h) How do you evaluate the trade-off between economic development and conservation objectives?

| Economic DST | SeaSketch |
|---|---|
| Uses of DST | Interactive mapping and visualization tool for collaborative MSP. |
| Country/MSP | SeaSketch has been employed in the Magallanes Region (Chile) and Massachusetts Ocean Management Plan (USA). |
| Limitation | The effectiveness of SeaSketch is dependent on the availability and quality of spatial data and active participation from stakeholders. |
| Data-intensive level | High-resolution data. Up-to-date datasets on marine ecosystems, habitats, species distributions, and socioeconomic factors such as tourism revenue. |

**Table A11.** *Cont.*

| Economic DST | SeaSketch |
|---|---|
| Output variable | Zoning plans, maps of potential impacts, cumulative impact assessments, and scenario-based visualizations. |
| References | [82]<br>https://legacy.seasketch.org/projects/ (14 February 2023) |

Source: Authors' creation based on literature reviewed.

**Table A12.** (i) How do you estimate the economic value of ecosystem services provided?

| Economic DST | Ecopath with Ecosim (EwE) Is a Software Package Used for Ecosystem-Based MSP |
|---|---|
| Uses of DST | Impact assessment. EwE can simulate the effects of human activities, such as fishing or climate change, on the marine environment and assess the potential ecological consequences. |
| Country/MSP | EwE has been utilized in South Africa for MSP, specifically in the Benguela Current region, to assess the impacts of different fishing scenarios on the ecosystem and inform management decisions. |
| Limitation | EwE requires extensive data on species interactions, population dynamics, fishing effort, and environmental variables. |
| Data-intensive level | High. EwE models require comprehensive data on species composition, trophic interactions, life history parameters, fishing effort, environmental variables (e.g., temperature, productivity), and other relevant ecosystem characteristics. |
| Output variable | Estimates of species biomass, trophic interactions, fishing impacts, and indicators of ecosystem health. These outputs can guide decision-making in MSP. |
| References | [83] |

Source: Authors' creation based on literature reviewed.

**Table A13.** (j) How do you assess possible conflicts among users of the ocean environment, and propose effective spatial management plans?

| Economic DST | Multi-Criteria Decision Analysis (MCDA) |
|---|---|
| Uses of DST | Evaluating and comparing multiple criteria and alternatives in decision-making. |
| Country/MSP | MCDA applied in New Zealand's MSP process to assess and prioritize different management scenarios based on ecological and socioeconomic criteria. |
| Limitation | Subjective weighting of criteria, stakeholder engagement challenges, and difficulty in quantifying certain criteria. |
| Data-intensive level | High: Data on ecological, economic, and social factors, stakeholder preferences, and decision criteria. |
| Output variable | Evaluation scores, ranking of alternatives, and trade-off analysis. |
| References | [84,85] |

Source: Authors' creation based on literature reviewed.

**Table A14.** (k) Governance: How do you incorporate new uses into MSP?

| Economic DST | Marine Spatial Planning Support System (MSPSS) |
|---|---|
| Uses of DST | MSPSS can be used to incorporate new uses, such as renewable energy or aquaculture. |
| Country/MSP | South Africa, Australia, Norway. E.g., The South African MSPSS aids in incorporating various uses, such as offshore oil and gas exploration or marine tourism, into the MSP process. |
| Limitation | Stakeholder engagement and local context are crucial for effective use of MSPSS, and its effectiveness may vary depending on the level of stakeholder involvement. |

**Table A14.** *Cont.*

| Economic DST | Marine Spatial Planning Support System (MSPSS) |
|---|---|
| Data-intensive level | Regional-scale datasets such as spatial data on ecological features (habitats, species distributions), socioeconomic factors (e.g., human activities), and governance aspects (e.g., regulations, management measures). |
| Output variable | Spatial allocation of different uses (e.g., areas offshore wind farms). Quantitative assessments of the impacts of uses on ecological, socioeconomic, and governance factors. |
| References | [86,87] |

Source: Authors' creation based on literature reviewed.

**Table A15.** (l) Governance: What are the most likely reactions of actors in marine space to the new regulations?

| Economic DST | Probabilistic Risk Assessment (PRA), Sensitivity Analysis Aiding Environment Risk Assessment |
|---|---|
| Uses of DST | Identifies the types of incentives needed to motivate higher compliance with regulations. Identifies and quantifies potential risks associated with marine activities, such as oil and gas exploration, shipping, or renewable energy installations. |
| Country/MSP | MSP scenarios in the German EEZ of the North Sea. Gulf of Mexico (USA) has employed PRA to evaluate risks from offshore energy development, spills, and natural hazards, and to inform decision-making processes for resource management. |
| Limitation | May not fully capture spatial and temporal variations in risks, particularly at finer scales or when considering dynamic changes in environmental conditions and human activities. |
| Data-intensive level | Data on hazards (e.g., pollution sources), exposure (e.g., spatial distribution of activities), vulnerability (e.g., ecological sensitivity, socio-economic factors), and potential impacts. |
| Output variable | Risk estimates, probability distributions, sensitivity indices, and visualizations of potential impacts and their uncertainties. These outputs guide the development of risk management strategies. |
| References | [66,88] |

Source: Authors' creation based on literature reviewed.

**Table A16.** (m) Governance: Is flexibility an explicit component of MSP?

| Economic DST | Adaptive Management (AM) |
|---|---|
| Uses of DST | AM helps reduce risks by allowing for adjustments and course corrections in response to unforeseen or unexpected outcomes. |
| Country/MSP | The Great Barrier Reef Marine Park (Australia), the Florida Keys National Marine Sanctuary (USA), and the Netherlands Integrated Management Plan for the North Sea. |
| Limitation | Adaptive management requires ongoing monitoring, data collection, analysis, and stakeholder engagement, which can be time-consuming and resource-intensive. |
| Data-intensive level | Data or information gathered through stakeholder engagement processes, such as surveys, interviews, or participatory mapping. |
| Output variable | Publication of new scientific knowledge, best practices, or guidelines that emerge from the adaptive management process. |
| References | [9] |

Source: Authors' creation based on literature reviewed.

**Appendix B. MSP Case Studies**

Early MSP initiatives are applied in marine conservation hotspots, and environmental objectives are prominent (e.g., the Great Barrier Reef in Australia). For the past two decades, MSP have been concentrated in areas where economic objectives are more paramount, with the aim of promoting sustainable development [89]. A case study approach is undertaken

to illustrate and contextualize how economic questions are incorporated in recent MSP. The examples and case studies, however, are by no means exhaustive. The following considerations are based on what is addressed in the current literature as well as what has emerged in grey literature. We have pointed out the importance of acknowledging context and recognizing jurisdictions similar to BC Canada. We pose some important economic questions which are related to MSP. The ways in which economic questions are integrated into MSP implementation are illustrated here in the case studies of the Gaufre Project in Belgium, South Africa, and South-East Australia.

*Appendix B.1. Case Study 1: Gaufre Project in Belgium*

The Gaufre Project is an MSP initiative undertaken by the Belgian government to efficiently manage its marine resources and ensure sustainable economic activities in the North Sea. The project, named after the Belgian waffle, aims to strike a balance between conservation efforts and the promotion of various economic activities within its EEZ. This case study explores the economic activities similar to BC in Canada, the conflicts that occurred during the planning process, the approaches used to resolve these conflicts, and the information needed for socio-economic planning and monitoring in the Gaufre Project.

The marine environment of Belgium's North Sea, like BC's coastal waters, supports a diverse range of economic activities. Some of the key activities similar to BC include the following: (1) Belgium's North Sea supports a significant fishing industry, just like BC, which relies on various species of fish and shellfish for both domestic consumption and export. (2) The coastal regions of Belgium attract tourists for beach activities, water sports, and seaside leisure, similar to BC's coastal tourism industry. (3) Both regions are interested in harnessing the potential of offshore renewable energy sources, such as wind farms, to reduce dependence on fossil fuels and mitigate climate change. (4) Belgium's North Sea hosts important shipping lanes and ports, handling trade and commerce similar to BC's port activities. (5) Conservation and protection of marine biodiversity and ecosystems are critical in both regions due to the ecological importance of their marine habitats.

During the MSP process, several types of conflicts emerged: (1) Spatial overlap conflicts arose when different economic activities, such as fishing grounds and offshore wind farm sites, overlapped, leading to competition for limited marine space. For instance, the application for a wind turbine construction and operation concession by Seanergy sparked a major conflict between offshore wind energy producers and a coalition of local inhabitants, coastal municipalities, and environmental NGOs [90]. These groups had differing visions for the use of marine space [90]. (2) Use vs. conservation conflicts: Another prominent conflict category arises from the tension between utilization and conservation. Activities that may require restrictions or even prohibition within certain marine areas, such as the approval of the Marine Environmental Protection Act, generated conflicts between the federal government and associations of fishermen, shipowners, recreational fisheries, and local politicians [91]. In response to these conflicts, the Belgian Government has adopted various approaches to facilitate resolution and ensure sustainable marine spatial planning: (1) Appointment of a Minister of the North Sea. In 2003 (and again in 2011), the government appointed a dedicated Minister of the North Sea with a specific mandate to coordinate all federal North Sea matters, excluding fisheries. This step aimed to streamline decision-making and improve governance in the maritime domain. (2) Emphasis on nature conservation. Growing awareness of the importance of nature conservation through EU Directives, such as the Marine Strategy Framework Directive (MSFD), has guided policymaking. These directives have encouraged the integration of environmental considerations into MSP processes. (3) Stakeholder engagement and transparency. The government has actively engaged stakeholders and the public throughout the MSP process. This approach has included transparent consultation rounds with major stakeholders, fostering trust, and reducing conflicts during the designation of MPAs between 2003 and 2005 [92]. For effective socio-economic planning and monitoring in the Belgian marine space, several key information needs must be addressed: (1) Advisory Committee on MSP:

The establishment of an advisory committee dedicated to MSP can help provide expert guidance and ensure that planning decisions align with sustainable development goals. (2) Legally binding MSP regulations. Belgium should continue to develop and enforce legally binding MSP regulations that specify permits and requirements for different marine activities. These regulations should be based on the principles of Good Environmental Status (GES) to safeguard the marine environment. (3) Data and monitoring. Robust data collection and monitoring systems are essential to assess the impacts of economic activities on the marine environment. This includes data on biodiversity, water quality, shipping traffic, and energy production. (4) Regular review and adaptation. MSP plans should be regularly reviewed and adapted to accommodate changing economic, environmental, and social conditions in the marine space.

The socio-economic information on the Belgian coastal area is similar to BC: The Belgian coastal region boasts a distinctive social milieu marked by a higher population density, seasonal employment opportunities, a substantial number of second homes, and elevated real estate prices. Furthermore, this area is home to significant economic hubs such as seaports and airports [93]. A key indicator for monitoring the MSP is the spatial productivity in the coastal area. Remarkably, these characteristics bear a resemblance to the British Columbia region.

*Appendix B.2. Case Study 2: MSP—Operation Phakisa, South Africa*

Operation Phakisa is an MSP initiative undertaken by the South African government to promote sustainable economic growth, conservation, and social development in the country's ocean space. Launched in 2014, the program seeks to address various challenges related to marine resource management and establish a coherent and coordinated approach to marine activities. This case study explores the economic activities similar to BC in Canada, the conflicts that occurred during the planning process, the approaches used to resolve these conflicts, and the information needed for socio-economic planning and monitoring in Operation Phakisa.

In the Phakisa Maritime Spatial Planning initiative, the economic activities are similar to those in British Columbia. They encompass aquaculture, marine protection and governance, offshore oil and gas, marine transport and manufacturing, coastal and marine tourism, and harbor development. These activities are essential for the region's economic growth and sustainability.

However, conflicts have arisen within the Phakisa Maritime spatial planning initiative. For instance, in late 2013, a significant conflict emerged involving Transnet, a primary investor. The national government's environmental officials rejected Transnet's Environmental Impact Assessment (EIA) for deepening the Durban port. This rejection occurred because Transnet had not adequately assessed the damage to the harbor's vital sandbank, which plays a critical role in preserving the ecosystem. Additionally, Transnet had not accurately evaluated the potential impacts of rising sea levels and severe storms [94].

To address conflicts such as these, the South African government employs various approaches. One approach involves engaging stakeholders with different levels of interest and knowledge, depending on the scale of their activities. By presenting data and plans at multiple scales, such as local intertidal areas for shore users or broader shoreline areas for boat-based fishers, discussions about how these activities affect each other and potential management actions can take place. This encourages dialogue, helps identify overlaps and conflicts, and fosters stakeholder engagement and alliances in broader planning issues. This approach aligns with the SeaPlan spatial process [95]. In terms of socio-economic planning and monitoring, the required information varies. Some aspects, such as project-specific or site-specific data, can be mapped relatively easily. Others, such as fishing and recreational activities, can vary in time and location. Key socio-economic information for Phakisa MSP includes coastal infrastructure, shipping casualties, outfalls (sewerage and stormwater), catchment management quality, fossil fuel mining, recreational oyster harvesting, commercial offshore crustacean trawling, and small-scale seine-net fishery

for sardines [95]. The operation of the Phakisa management team has outlined a set of performance indicators such as inclusive growth, job creation, enabling regulatory environment, funding support, increasing skills pool, and improving access to markets to help monitor delivery [96].

*Appendix B.3. Case Study 3: MSP in Australia*

Australia's MSP is a dynamic and inclusive process that strives to achieve sustainable economic development, environmental conservation, and social well-being. Through collaboration, data-driven decision-making, and innovative zoning strategies, MSP in Australia serves as a model for other regions, including BC, seeking to effectively manage their marine resources and balance diverse interests in a changing marine environment. This case study illustrates how socio-economic activities have been incorporated into various stages of MSP in terms of policy mechanisms, data requirements, data-sharing mechanisms, and replicability within a BC context.

Regarding the policy mechanisms in Australia, MSP integrates economic questions through a combination of federal, state, and territory-level policies. For example, the Great Barrier Reef Marine Park Authority integrates fisheries management, tourism regulations, and conservation policies to protect the reef while supporting sustainable economic activities [4].

Economic data required in Australia's MSP include tourism revenue, commercial fishing catches, and marine transport data. Data are collected from various government agencies, tourism boards, and industry bodies [6]. Data sharing mechanisms in Australia's MSP encourage data sharing through collaboration between federal and state governments, research institutions, and stakeholders [72]. The Marine Cadastre, an online spatial data portal, facilitates data sharing and accessibility.

In terms of replicability, BC can adopt Australia's multi-level policy approach and integrate economic activities into MSP through collaboration between federal and provincial authorities. Developing a centralized data portal and promoting data sharing among stakeholders will facilitate informed decision-making in BC's MSP process.

Economic questions play a crucial role in MSP, and integrating socio-economic activities into different stages of the MSP process is essential for achieving sustainable development. By implementing clear policy mechanisms, establishing data-sharing mechanisms, and using comprehensive socio-economic data, jurisdictions such as Australia, Norway, and Germany have effectively balanced economic activities with environmental and social considerations. BC can draw lessons from these case studies and replicate successful approaches to enhance its MSP process and promote sustainable economic development in its marine environment.

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
