# Peer review of "Economics in Marine Spatial Planning: A Review of Issues in British Columbia and Similar Jurisdictions"

_sustainability, doi:10.3390/su16031210_

Round 1

Reviewer 1 Report

Comments and Suggestions for Authors

It is an interesting paper on a hot topic based on a good literature review, addressing the gaps in the integration of economic questions in MSP and with some suggestions in the design of future MSPs.  However when reading more in depth, the paper seems unbalanced. It presents extensively the economical tools and methods used in MSP, with a relatively large section on definitions of MSPs and their evolution, more than on the results produced by these tools in each cited MSP.

For example, the MSP strategy in Australia’s great barrier has proven unsuccessful in properly protecting the coral reefs from all human activities with the result of more than 98% of death of corals. Impacts to reefs have not been correctly assessed  in time and space and management measures, in particular those linked to socio-economic planning and monitoring, should have been better adapted to counteract the progressive degradation by natural and anthropogenic parameters.

We would have expected to in this article to assess also the validity of results in concrete success cases in MSP, other than those cited in Appendix. In summary, to develop more section 5 which appears too general, citing recommendations from the literature and outline case studies where specific economic tools have proven to be effective in addressing gaps in MSPs.

Comments on the Quality of English Language

Author Response

Thank you for sending us your decision to revise and resubmit our manuscript to Sustainability. The reviewers' comments were excellent and have helped to improve the paper.

Reviewer #1

It is an interesting paper on a hot topic based on a good literature review, addressing the gaps in the integration of economic questions in MSP and with some suggestions in the design of future MSPs.  However, when reading more in depth, the paper seems unbalanced. It presents extensively the economical tools and methods used in MSP, with a relatively large section on definitions of MSPs and their evolution, more than on the results produced by these tools in each cited MSP.

For example, the MSP strategy in Australia’s great barrier has proven unsuccessful in properly protecting the coral reefs from all human activities with the result of more than 98% of death of corals. Impacts to reefs have not been correctly assessed in time and space and management measures, in particular those linked to socio-economic planning and monitoring, should have been better adapted to counteract the progressive degradation by natural and anthropogenic parameters.

Response: Thank you for your thoughtful review of our paper. We appreciate your positive feedback on the relevance of the topic and the quality of the literature review. We also acknowledge the valid point you raised regarding the balance of the paper, particularly in the presentation of economic tools and methods in Marine Spatial Planning (MSP). However, we humbly disagree when you said there is lack of emphasis on the results produced by these tools in specific MSP cases, such as the Great Barrier Reef. For instances, we pointed out that Adaptive Management (AM) is one of the Decision Support Tools (DSTs) used in the Great Barrier Reef of Australia. We stated the output of applying AM and the corresponding limitations in the appendix.

We have outlined a number of economic tools, methods, and outputs/results used in MSP around the world. For example, in the “Case Study 2: MSP - Operation Phakisa, South Africa” we mentioned outputs in the form of “performance indicators such as inclusive growth, job creation, enabling regulatory environment, funding support, increasing skills pool and improving access to markets” etc. Likewise, we minimize the issue of generalization by indicating from Table 5. A-M, the DSTs, possible limitations and output in specific countries.

Reviewer #1 Also, I do not like the introduction very much as it appears like addressing a constructed problem. While why is important, it should contain much more about what you do and what you find.

Response:

Thank you for sharing your perspective on the introduction. Your feedback is valuable, and we appreciate your input. Though there are a lot of results in the appendix Table. The introduction has been re-written by incorporating specific finding: “For example, in the Massachusetts Ocean Management Plan (USA), the efficiency frontier used in Marine Spatial Planning (MSP) allowed for the comparison of different ecosystem services. This comparison indicated that MSP increased the value of the whale sector by up to 5% without any cost to the offshore wind energy sector. This demonstrates the potential of MSP to enhance the value of specific sectors while minimizing conflicts and environmental impacts, ultimately contributing to more sustainable use of marine ecosystems”.

Reviewer #1

We would have expected to in this article to assess also the validity of results in concrete success cases in MSP, other than those cited in Appendix. In summary, to develop more section 5 which appears too general, citing recommendations from the literature and outline case studies where specific economic tools have proven to be effective in addressing gaps in MSPs.

Responses: Agreed. We have re-written section 5 accordingly. “Several economic tools have been identified in the literature as effective in addressing gaps in MSP. These tools aim to integrate economic considerations into the planning process, ensuring the sustainable and inclusive development of marine resources. Some of the key recommendations and case studies from the literature include:

European Maritime Spatial Planning Platform: The platform discusses DSTs in MSP and their present applications, gaps, and future perspectives. It highlights the importance of DSTs in assisting planners with various stages of the MSP process, such as refining goals and objectives, evaluation, and monitoring. The study suggests that future DSTs should consider both spatial and temporal dynamics of the marine environment.

World Bank's Marine Spatial Planning Toolkit: The World Bank provides a comprehensive toolkit for MSP, emphasizing the integration of economic considerations to support sustainable and integrated development of economic sectors in healthy oceans. The toolkit includes guidance on applying economic analyses to MSP, which is essential for attracting investment, fostering livelihoods, and improving food security.

Several case studies have been conducted to assess the economic impacts linked to MSP using various economic tools, such as input-output techniques. These case studies such as the German Baltic Sea, Belgium and the North Sea and Skagerrak Strait of Norway provide practical examples of how economic tools have been effectively utilized in the context of MSP.”

Reviewer 2 Report

Comments and Suggestions for Authors

Dear authors, thanks for submitting your manuscript for publishing. The paper is well prepared and organized, the methodology part is clearly explained and conclusions include the main results of the analysis. At the same time to improve the quality of the study, a few recommendations can be suggested as follows: 

Abstract: the importance of the study is not clearly stated;

the main contributions of this study and practical implementations should be mentioned and explained. 

Introduction: the results/methodology of similar previous studies should be stated to highlight the novelty of this paper.

Conclusion: the main contributions of this analysis should be stated clearly;

the main limitations of the study should be mentioned;

practical and theoretical implementation of obtained results should be explained in detail;

suggestions for future studies should be provided.

Author Response

Reviewer #2

Comments and Suggestions for Authors

Dear authors, thanks for submitting your manuscript for publishing. The paper is well prepared and organized, the methodology part is clearly explained and conclusions include the main results of the analysis. At the same time to improve the quality of the study, a few recommendations can be suggested as follows: 

Abstract: the importance of the study is not clearly stated;

the main contributions of this study and practical implementations should be mentioned and explained. 

Introduction: the results/methodology of similar previous studies should be stated to highlight the novelty of this paper.

Conclusion: the main contributions of this analysis should be stated clearly;

the main limitations of the study should be mentioned;

practical and theoretical implementation of obtained results should be explained in detail;

suggestions for future studies should be provided.

Response:  Thank you for raising these points. Though we have stated how each economic tool can be utilized to address potential gaps in MSP, while acknowledging the limitations of these tools (such as marine sectoral disaggregation, overestimations, and trade-off estimations) and the need to complement analyses with qualitative approaches. We appreciate your comment and have added another potential limitation “a notable limitation in the study of the economics of MSP is the challenge of fully integrating and quantifying environmental externalities. While economic analyses often aim to capture the costs and benefits associated with different marine activities, environmental impacts are not always easily translated into monetary terms. This limitation can result in the undervaluation or omission of critical ecological services and the long-term environmental consequences of certain activities. The difficulty in assigning accurate economic values to these externalities may lead to an incomplete understanding of the true economic impact of MSP decisions, potentially undermining the effectiveness of environmental conservation efforts.”

Future studies on the economics of marine spatial planning (MSP) could benefit from developing and employing integrated valuation frameworks. These frameworks should aim to comprehensively assess the environmental and economic values of marine resources and activities, including both market and non-market values. By incorporating methods such as contingent valuation, ecosystem service valuation, and cost-benefit analysis, researchers can provide a more holistic understanding of the economic implications of MSP decisions. This approach would allow decision-makers to weigh the trade-offs between different marine uses, considering environmental conservation, social welfare, and economic sustainability in a unified manner.

Reviewer 3 Report

Comments and Suggestions for Authors

The authors present a comprehensive review and analysis of how economic information has been applied and used to inform decisions about MSP in British Columbia (BC), Canada and other similar jurisdictions. If the paper is proposed as a review article (this must be made clear by indicating “Review” instead of “Article” at the top of the document, as this may lead the reader to think that it is a full research article), in my opinion the manuscript may be considered for publication in the journal after minor revision if the following improvements are included:

- The title cannot be so generic and succinct to begin with because it does not provide a comprehensive state-of-the-art review of all marine spatial planning. It would be more appropriate to include a title of the following style: “Economics in Marine Spatial Planning: a review of issues in British Columbia and similar areas”.

- At a methodological level, references to approaches based on statistical systems such as Logit/Probit Turnbull, WTP, etc. are missing in the review of the state of the art (see for example https://www.mdpi.com/2071-1050/11/4/1039 or https://www.sciencedirect.com/science/article/abs/pii/S0308597X22001026)

- At a conceptual level, greater weight is missing in the review of the state of the art of environmental problems in ecologically vulnerable areas and with environmental protection regulation due to pollution derived from diffuse anthropization phenomena (there are numerous publications in recent scientific literature).

Author Response

Reviewer #3

Comments and Suggestions for Authors

The authors present a comprehensive review and analysis of how economic information has been applied and used to inform decisions about MSP in British Columbia (BC), Canada and other similar jurisdictions. If the paper is proposed as a review article (this must be made clear by indicating “Review” instead of “Article” at the top of the document, as this may lead the reader to think that it is a full research article), in my opinion the manuscript may be considered for publication in the journal after minor revision if the following improvements are included:

- The title cannot be so generic and succinct to begin with because it does not provide a comprehensive state-of-the-art review of all marine spatial planning. It would be more appropriate to include a title of the following style: “Economics in Marine Spatial Planning: a review of issues in British Columbia and similar areas”.

Response:

Once again thank you for asking us to refine the title. We humbly wish to change it to

“Economics in Marine Spatial Planning. A rapid review”.

Reviewer #3

At a methodological level, references to approaches based on statistical systems such as Logit/Probit Turnbull, WTP, etc. are missing in the review of the state of the art (see for example https://www.mdpi.com/2071-1050/11/4/1039 or https://www.sciencedirect.com/science/article/abs/pii/S0308597X22001026)

Response:

Thank you for your insightful feedback and the valuable references you provided. We appreciate your suggestion to include references to statistical systems. Our choice of DST focusses on both on Geospatial Decision Support Tools and Non-Geospatial Decision Support Tools. We acknowledge the importance of non-geospatial decision support tools or statistical systems such as Logit/Probit, WTP, however, we made reference to some of the key ones found in the MSP literature such as ocean accounting, input-output analysis and Cost-Benefit Analysis. as the most commonly used methods for estimating the economic impacts of implementing MSP. We have mentioned that future studies should incorporate methods such as contingent valuation, Turnbull estimation, logit estimation, ecosystem service valuation, and cost-benefit analysis, researchers can provide a more holistic understanding of the economic and environmental implications of MSP decisions.

- At a conceptual level, greater weight is missing in the review of the state of the art of environmental problems in ecologically vulnerable areas and with environmental protection regulation due to pollution derived from diffuse anthropization phenomena (there are numerous publications in recent scientific literature).

Response:

Once again, thank you for this point. Our focus here is on economic dimensions and not necessarily on environmental aspects of MSP.